# Therapeutic effect of T-cell engager in two patients with autoimmune neuropathy

Jonathan Wickel [1,6], Mihai Ceanga[1,6], Benjamin Vlad [1], Nikolai von Stackelberg [1], Nounagnon Romaric Tochoedo[2], Danilo Schulz[2], Olaposi Yomade[3], Diana Dudziak [2,4,5] & Christian Geis [1] ✉

Chronic immune-mediated peripheral nerve myelinopathies (CIPNM) are progressive and debilitating disorders that can be refractory to current treatment regimes. The bispecific T-cell engager (BiTE) teclistamab depletes B-cell Maturation Antigen (BCMA) positive late-stage B- cells and plasma cells by engaging T-cells. We present two patients with treatment-refractory CIPNM (Patient 1: IgM-kappa-associated CIPNM, follow-up 9 months; Patient 2: Anti-myelin-associated-glycoprotein (MAG) antibody-mediated CIPNM, follow-up 6 months). Both patients are characterized by rapid improvement upon treatment with teclistamab: substantially increased walking distance, accompanied by reduced nerve swelling and markedly improved electroneurography. Furthermore, some nerves without stimulus response at baseline are showing detectable response during follow-up. The functional recovery of peripheral nerves is paralleled by decreased serum neurofilament levels, indicating reduced neuronal damage. IgM kappa paraprotein and high-titer MAG antibodies are undetectable 6 weeks after the first dose of teclistamab and remain negative throughout the follow-up, whereas soluble BCMA re-emerges following an initial reduction. No serious adverse events are observed during treatment. This case series highlights the significant therapeutic potential of teclistamab in treatment-refractory CIPNM and provides an important example of 'off-the-shelf' BiTE therapy being well tolerated and effective and should be further investigated in neuro-immunological disorders.

Recent reports showed promising efficacy of chimeric antigen receptor (CAR) T-cell therapy in neuroimmunological diseases[1-4]. Building upon these novel developments in neuroimmunology, we here report the first patients treated with a bispecific T-cell engager (BiTE) for therapy-refractory chronic immune-mediated neuropathies. BiTEs are therapeutic antibody constructs that connect patient's effector T-cells to specific target cells, such as B-cells through a CD19 binding domain or plasma cells via a B-cell maturation antigen (BCMA). Similar to CAR T-cell therapy, this interaction leads to rapid B-cell and plasma cell depletion through T-cell-mediated cytotoxicity thereby reducing pathogenic antibody titers.

Chronic immune-mediated peripheral nervemyelinopathies (CIPNM) are progressive and debilitating disorders that include typical chronic inflammatory demyelinating polyradiculoneuropathy (tCIDP)

[1]Section Translational Neuroimmunology, Department of Neurology, Jena University Hospital, Jena, Germany. [2]Institute of Immunology, Friedrich Schiller University, Jena University Hospital, Jena, Germany. [3]Klinik für Innere Medizin II, Abteilung für Hämatologie und Internistische Onkologie, Jena University Hospital, Jena, Germany. [4]Cluster of Excellence Balance of the Microverse, Friedrich Schiller University Jena, Jena, Germany. [5]Core Facility Cytometry, Jena University Hospital, Jena, Germany. [6]These authors contributed equally: Jonathan Wickel, Mihai Ceanga. ✉e-mail: christian.geis@med.uni-jena.de

and multifocal (mf)CIDP, CIPNM with nodal/paranodal serum antibodies (nodopathies), and neuropathies associated with IgM paraproteins, e.g., anti-myelin-associated glycoprotein antibodies (anti-MAG-Ab; CIPNM-MAGAb), monoclonal IgM, or light chain subtypes[5]. Most patients with CIDP subtypes and CIPNM with nodal/paranodal serum antibodies respond well to treatments such as corticosteroids, plasma exchange, or intravenous/subcutaneous immunoglobulins (IVIG/SCIG)[6]. However, patients with neuropathies linked to IgM paraproteins often have inadequate responses to therapy, including B-cell depletion with rituximab, or may only experience delayed treatment effects[7,8]. In CIPNM-MAGAb, a sustained antibody titer reduction of at least 50% has been proposed as a marker of clinical treatment response[9].

Here, we applied teclistamab, a BiTE approved for the treatment of relapsed or refractory multiple myeloma[10], as a first-in-class treatment for neuroimmunological diseases. Teclistamab was recently shown to efficiently reduce disease activity in systemic autoimmune disease[11–13] and in a patient with a rare subtype of myasthenia gravis[14]. It binds to CD3 and BCMA, resulting in rapid T-cell-mediated depletion of late-stage B-cells and plasma cells.

This study evaluates teclistamab in two patients with treatment-resistant chronic immune-mediated peripheral nerve myelinopathies (CIPNM). Both show rapid improvement in symptoms, including better walking ability and reduced nerve damage, with no serious side effects. Teclistamab effectively depletes paraproteins and MAG antibodies.

The findings suggest teclistamab as a promising treatment for refractory CIPNM, warranting further exploration in neuroimmunological disorders.

## Results

Both patients presented with progressive, treatment-refractory autoimmune neuropathy associated with either IgM-kappa paraproteins or MAG-Abs and were at risk of losing ambulatory function.

### Patient 1

A 65-year-old woman with a five-year history of IgM-kappa paraproteinemic neuropathy presented with progressive distal sensory loss in the arms and legs, leg and lower back pain, and gait disturbance. Despite previous treatments, including high-dose steroids, IVIG and plasma exchange, her clinical status progressively deteriorated (Supplementary Table 1) and she presented with a reduced maximal walking distance (baseline: 300 meters). She typically necessitated bilateral support walking outdoors, exhibited significant difficulty climbing stairs, and was unable to run or tolerate prolonged standing. As nerve conduction studies demonstrated a sensory-motor neuropathy with marked demyelinating features, while sonography showed pronounced nerve swellings, she was diagnosed with CIPNM and IgM-kappa paraproteins.

Teclistamab induced a rapid clinical response, marked by increased walking distance, enhanced grip strength, and improved neuropathy assessment scores (INCAT, I-RODS) along with significant recovery in nerve conduction velocity, distal motor latency, and compound muscle action potential amplitudes, as well as resolution of nerve swelling. Patient-reported outcomes, including quality of life, pain, and fatigue also improved (Fig. 1, SFig. 1, Supplementary Table 2). In good agreement with the clinical and neurophysiological improvement, serum neurofilament light chain (Nfl) levels also continuously decreased upon treatment with teclistamab, indicating an attenuation of neuronal injury (SFig. 1C). Peripheral blood flow cytometry analysis revealed a transient consumption of T-cells followed by a depletion of B-cells with B-cell repopulation starting approximately four months after teclistamab treatment (Fig. 2A). Repopulating B-cells were predominantly characterized by a naïve phenotype at month 6 and 9 whereas plasmablasts and plasma cells as well as more differentiated B-cells were barely detectable in peripheral blood and did not change

substantially (Fig. 2B, SFig. 2). These findings are in line with previous reports on BiTE and CAR T-cell treatment in autoimmune diseases targeting CD19[15,16]. Concurrently, soluble BCMA (sBCMA), a short half-life biomarker of late B-cells, plasmablasts, and plasma cells[17], also decreased rapidly and reoccurred before re-expansion of circulating B-cells post-treatment (Fig. 2C). Interestingly, M-Protein completely disappeared and was undetectable until last follow-up at 9 months (day 287), possibly indicating a polyclonal rather than monoclonal re-expansion of plasmablasts and plasma cells following teclistamab treatment. In line with this, six months after teclistamab treatment, kappa free light chains became detectable again despite the continued absence of M-protein (Fig. 2C). Levels of IgM, IgG and IgA also decreased, while vaccination titers were largely preserved (Fig. 2D, SFig. 3). Adverse events were mild, including a grade 2 cytokine release syndrome (CRS), neutropenia, and hypogammaglobulinemia, which were well controllable with a single administration of tocilizumab and granulocyte colony-stimulating factor (G-CSF), and two IVIG infusions, respectively (Fig. 2E). The positive clinical and paraclinical response was maintained without signs of disease reoccurrence until the last follow-up (month 9).

### Patient 2

A 73-year-old woman with a 13-year history of progressive sensory-ataxic gait and distal-predominant sensorimotor impairment was diagnosed with CIPNM associated with IgM-kappa paraproteins. After initial improvement of sensory deficits on high-dose steroid therapy, her disease course was progressive over the years despite multiple treatments including steroids, IVIG, plasma exchange, and rituximab, resulting in ataxic gait and distal pareses of the lower extremities with recurrent falls and fractures (Fig. 3A; Supplementary Table 1). Nerve conduction studies showed loss of motor and sensory responses in leg nerves, with profound conduction slowing of the arm nerves, and multifocal nerve swelling on ultrasound. MAG-Abs were detected, establishing the diagnosis of CIPNM-MAGAb.

Teclistamab treatment increased walking distance over sixfold within 6 months and improved clinical scores (i.e., grip strength, I-RODS, MRC sum, INCAT, Timed Up and Go test) and nerve conduction studies (Fig. 3A, SFig. 4A-B, Supplementay Table 3). Ultrasound confirmed a marked reduction of nerve swelling (Fig. 3A, SFig. 4C). The patient also reported improvement in quality of life as well as reduced neuropathy impairment and pain (Fig. 3B). Teclistamab induced intermittent lymphopenia with a transient drop of CD3 + T-cells and complete depletion of B-cells (Fig. 4A). Again, sBCMA levels declined rapidly after treatment, followed by an increase three months after teclistamab. Remarkably, similar to IgM-kappa paraprotein levels in patient 1, M-Protein was not detectable until last follow up (Fig. 4B). Serum Nfl levels were also decreased (Fig. 4C). MAG-Ab titers decreased within the first month of therapy and remained negative until last follow up at 6 months (day 186) (Fig. 4D). Serological analysis showed a reduction of total IgM and IgG levels. Except for tetanus IgG, vaccine titers remained largely intact (SFig. 3). Following the first and second administration, the patient experienced grade 1 CRS with moderate IL-6 increase. In contrast to patient 1, no neutropenia was observed (Fig. 4F). As expected, she developed hypogammaglobulinemia, for which she received IVIG infusions (Fig. 4E). At month 5, the patient developed mild bronchitis, accompanied by moderate increases in IL-6 and neutrophil levels (Fig. 4F), without the need for antibiotic therapy or hospitalization. Clinical and paraclinical improvement remained stable with no signs of relapse until the last follow-up (month 6).

## Discussion

Administration of four teclistamab doses resulted in rapid and significant improvement of neuropathy-induced motor and non-motor symptoms in two previously treatment-refractory patients. Notably,

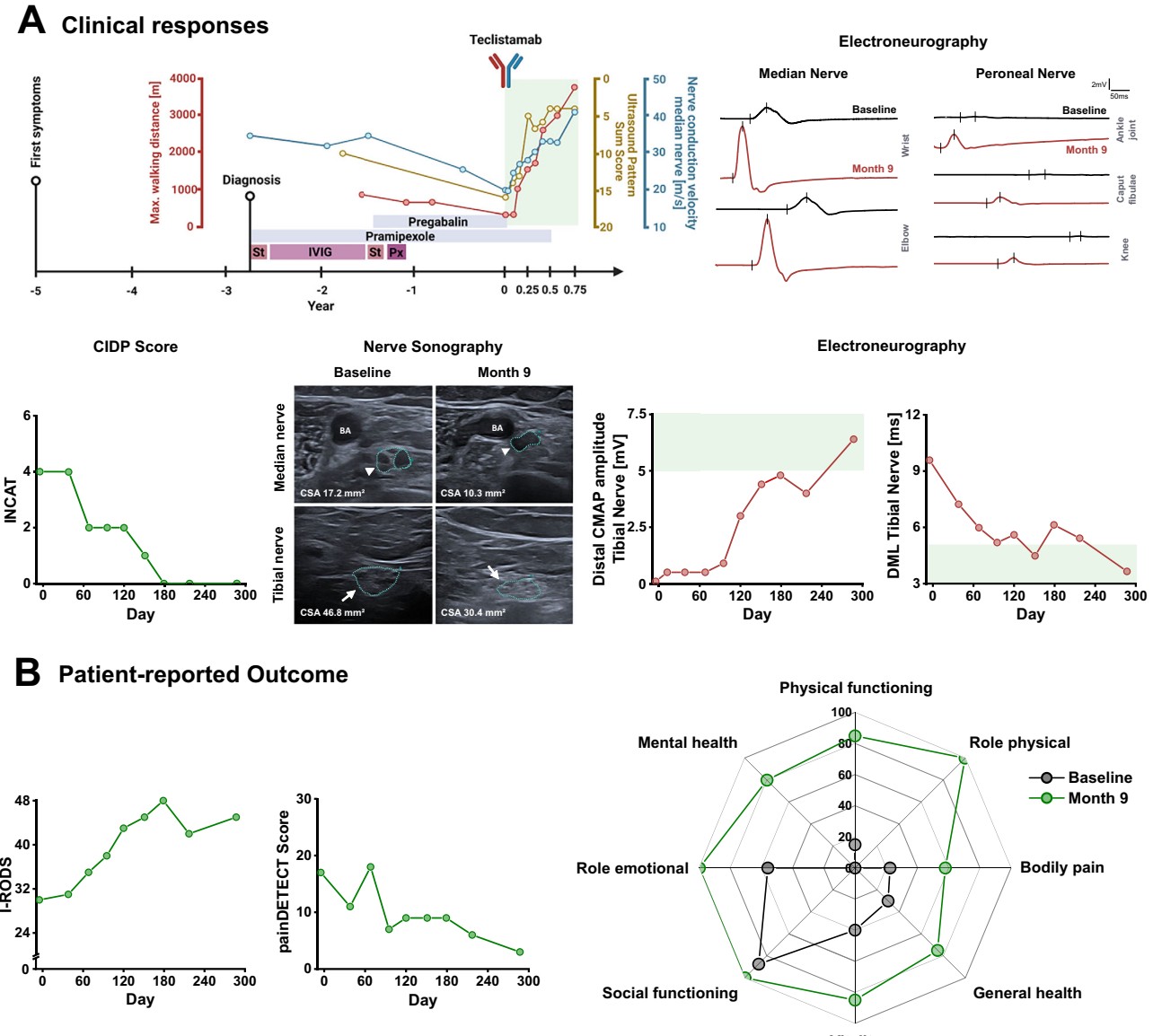

**Fig. 1 | Patient 1: Clinical responses and patient-reported outcome in refractory IgM-paraproteinemic autoimmune neuropathy treated with teclistamab.**
**A** Upper Panel: Left: Overview of disease course until 9 months (287 days) post-treatment follow up, prior treatment approaches, ultrasound pattern sum score (UPSS; both sides; range 0 [normal] to 44 [most severe]), and nerve conduction velocity recordings [right median nerve; normal value: > 50 m/s]. Green area: Results after teclistamab treatment. Right: Electroneurographic recordings of right median and left peroneal nerve prior treatment with teclistamab (baseline; black) and 9 months after first teclistamab dose (red) showing increased amplitudes and reduced distal motor latency (DML). Lower Panel: Left: Inflammatory Neuropathy Cause and Treatment disability score (INCAT; range 0 [normal] to 10 [most severe]) illustrating improvement of neuropathy symptoms. Nerve sonography: Median (arrow-head) and tibial nerve (arrow) sonography at baseline and month 9 showing a reduction of nerve swelling. Right: Distal compound muscle action potential (CMAP) amplitude of the right and DML recordings of the left tibial nerve showing a profound improvement. Green area: Normal values. **B** Patient-reported Outcome: Inflammatory Rasch-built Overall Disability Scale (I-RODS) is a patient-reported outcome scale specifically designed to assess activity and social participation limitations in patients with inflammatory polyneuropathies (range 0 [most severe] to 48 [normal]). painDETECT score: painDETECT is a patient-completed screening questionnaire designed to identify neuropathic pain components in individuals with chronic pain conditions (range 0 [no pain] to 35 [most severe]). The Short Form 36 Health Survey (SF-36): SF-36 at baseline and month 9. The SF-36 is a widely used, patient-reported questionnaire designed to measure health-related quality of life across a broad range of diseases and populations. Abbreviations: St: high-dose steroids, IVIG: intravenous immunoglobulins, Px: plasma exchange, CIDP: Chronic inflammatory demyelinating polyneuropathy, BA: brachial artery, CSA: cross-sectional area CMAP: compound muscle action potential, DML: distal motor latency. Source data are provided as a Source Data file.

this clinical recovery was accompanied by a swift and marked improvement in nerve function, particularly electrophysiological indicators of myelinopathy and serological evidence of reduced neuronal injury. Collectively, these findings support the direct pathogenic role of MAG-Abs and IgM paraprotein in CIPNM as previously reported[8,18,19]. Although patients may also respond to depletion of circulating B-cells using rituximab, the reduction of IgM is often incomplete and delayed, resulting in an insufficient therapeutic response[9]. Targeting BCMA, expressed on late-stage B-cells, plasmablasts and plasma cells, using teclistamab enables a presumably deeper reduction of pathogenic paraproteins than conventional treatment regimes, e.g., rituximab. These observations further support the concept that CIPNMs associated with IgM paraproteins are monoclonal gammopathies of neurological significance (MGNS) and underline the

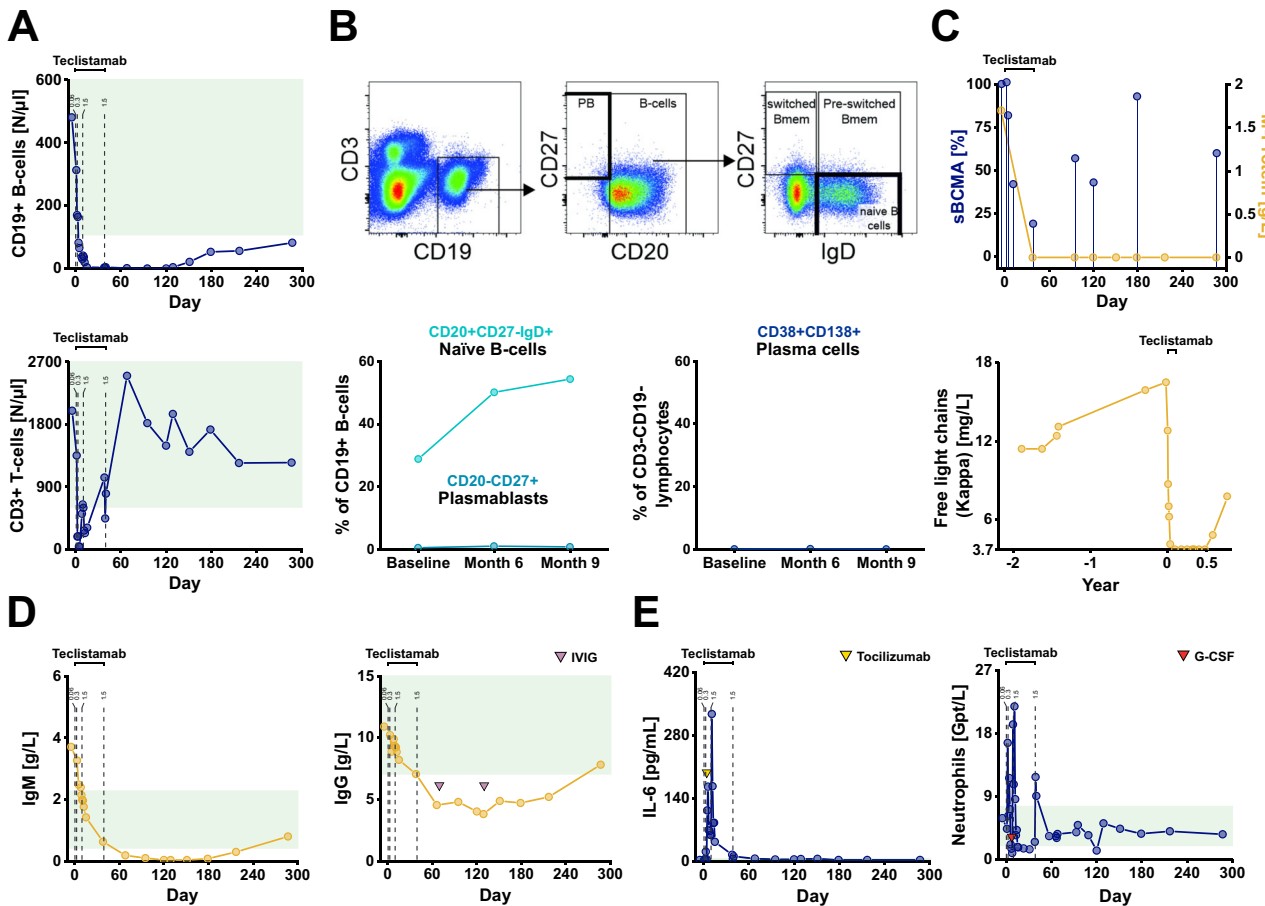

**Fig. 2 | Patient 1: Immunological changes and serologic responses in refractory IgM-paraproteinemic autoimmune neuropathy treated with teclistamab.**
**A** Course of absolute CD19 + B-cell and CD3 + T-cell counts illustrating depletion and repopulation of CD19 + B-cells following teclistamab therapy. Green area: Normal values. **B** Fluorescence-Activated Cell Sorting Analysis of naïve B-cells, plasmablasts and plasma cells before and after teclistamab treatment using peripheral blood. Upper panel: excerpt of the gating strategy for naïve B-cells and plasmablasts. Lower Panel: course of naïve B-cells, plasmablasts and plasma cells (time points: baseline/d0 prior teclistamab, month 6/d179 and month 9/d287). FACS analysis was performed once at the respective time point. **C** Relative sBCMA levels showed a temporary decrease during the treatment phase whereas the

M-protein remained undetectable until last follow-up (day 287). Free light chain (Kappa) levels were below the lower detection limit (3.7 mg/L) and increased again after approximately 7 months (day 217). sBCMA measurements were performed three times at each time point. **D** IgM and IgG were reduced after teclistamab treatment with subsequent increase. Purple triangles indicate time points of IVIG infusion. Green area: Normal values. **E** Interleukin-6 (IL-6) and neutrophil levels. Cytokine release syndrome was treated with tocilizumab (yellow triangle), neutropenia with granulocyte colony stimulating factor (G-CSF) (red triangle). Dashed lines in **A**, **D**, and **E** indicate time points of teclistamab injections (0.06: 0.06 mg/kg body weight; 0.3: 0.3 mg/kg body weight; 1.5: 1.5 mg/kg body weight). Source data are provided as a Source Data file.

importance of rapid and thorough IgM paraprotein reduction by depletion of the pathogenic plasma cell population. In patient 2, MAG-Abs were negative already 6 weeks after teclistamab treatment despite a high baseline titer (>1:1000). Prior treatment with rituximab produced neither a decrease in MAG-Ab titers nor any clinical benefit. Interestingly, although sBCMA rose again after an initial teclistamab-induced decline, the pathogenic MAG-Abs and IgM paraprotein remained undetectable. Together with the more naïve B-cell phenotype observed in the repopulating B-cell compartment, this data suggests that the newly emerging population of B-cells, plasmablasts, and plasma cells is predominantly polyclonal rather than reflecting a relapse of the previous monoclonal population. As expected, plasmablasts and plasma cells were hardly detectable in peripheral blood. It should be noted that repeated bone marrow or lymph node biopsies were not conducted in these compassionate use settings due to ethical considerations, which precluded detailed assessment of plasmablasts and plasma cells. A limitation of current findings is that they are derived from individual cases, and follow-up is currently limited to 9 months. Nevertheless, this post-treatment observation period is in good accordance to the time to the primary endpoint in many

randomized trials of autoimmune neuropathies[20–24]. Overall, teclistamab was well tolerated, with manageable side effects such as mild CRS, hypogammaglobulinemia, and neutropenia. Treatment using BiTEs may therefore provide an effective and straightforward therapeutic option in refractory CIPNM associated with pathogenic MAG-Abs and IgM paraprotein. In recent years, autologous antiCD19-/anti-BCMA-CAR T-cell therapy showed promising first results in several B-cell driven and autoantibody-associated diseases including myasthenia gravis, Lambert-Eaton myasthenic syndrome, Stiff-person syndrome, CIDP, and Neuromyelitis optica spectrum disorders, respectively[1–4]. Although the treatment is generally well tolerated in non-oncology patients and the concerns of an increased incidence of Immune Effector Cell-Associated Neurotoxicity Syndrome (ICANS) in neurological patients have not materialized, autologous CAR T-cell therapy is still limited to individual cases due to cost-intensive and potentially harmful procedures such as leukapheresis and lymphodepletion, as well as the risk of CAR-T lymphomas. In contrast, BiTEs are engineered therapeutic antibodies that are available 'off-the-shelf', do not require leukapheresis nor lymphodepletion, and allow for adjustable dosing based on clinical need. Our data and recent reports of BiTE therapy in

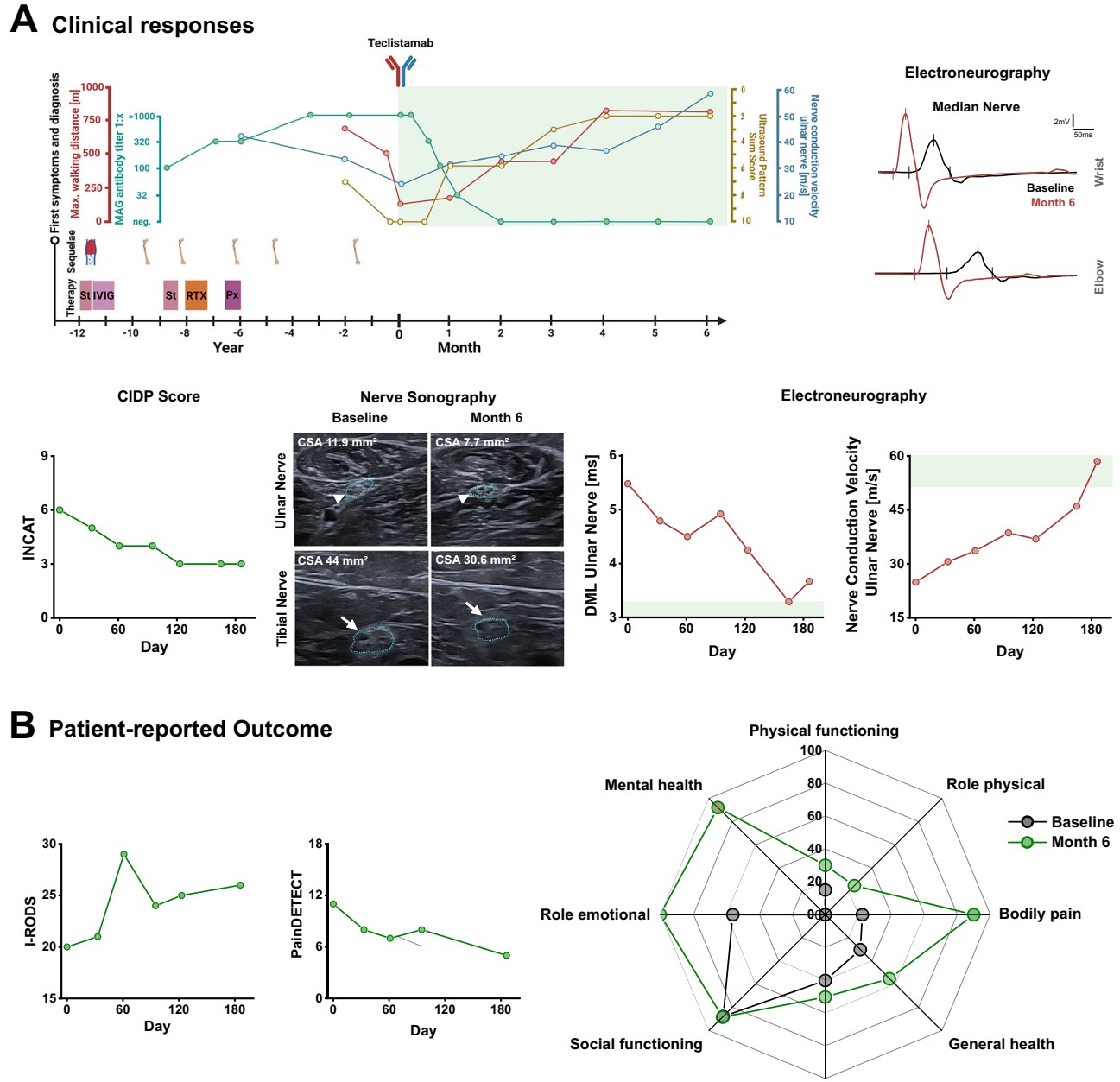

**Fig. 3 | Patient 2: Clinical responses and patient-reported outcome in refractory MAG neuropathy treated with teclistamab. A** Upper Panel: Left: Overview of disease course until follow-up 6 months (186 days) post-treatment, prior treatment approaches, sequelae of treatment or neuropathy, ultrasound pattern sum score (UPSS; right side; range 0 [normal] to 22 [most severe]), nerve conduction velocity recordings [left ulnar nerve; normal value: > 50 m/s], and MAG-Ab titers. Green area: Results after teclistamab treatment (note, change of x-axis intervals from years to months after teclistamab treatment). Right: Electroneurographic recordings of left median nerve prior treatment with teclistamab (baseline; black) and 6 months after first teclistamab dose (red) showing increased amplitudes and improved distal motor latency (DML). Lower Panel: Left: Inflammatory Neuropathy Cause and Treatment disability score (INCAT; range 0 [normal] to 10 [most severe])

illustrating improvement of neuropathy symptoms. Nerve sonography: Ulnar (arrow-head) and tibial nerve (arrow) sonography at baseline and month 6 showing a reduction of nerve swelling. Right: DML and nerve conduction velocity recordings of the left ulnar nerve showing a profound improvement. Green area: Normal values. **B** Patient-reported Outcome: Repetitive I-RODS (range 0 [most severe] to 48 [normal]) and painDETECT (range 0 [no pain] to 35 [most severe]) scores before and after teclistamab therapy. SF-36 at baseline and month 6 measuring health-related quality of life. Abbreviations: St: high-dose steroids, IVIG: intravenous immunoglobulins, RTX: rituximab, Px: plasma exchange, CIDP: Chronic inflammatory demyelinating polyneuropathy, CSA: cross-sectional area, CMAP: compound muscle action potential, DML: distal motor latency. Source data are provided as a Source Data file.

two patients with myasthenia gravis have shown their potential in severely affected patients with disorders of the peripheral nervous system[14,25]. Moreover, the library of bioengineered BiTE constructs will further increase eventually enhancing efficacy, target specificity, and blood-brain barrier permeability also for application for central nervous system immunopathology. In a larger context, these advantages

could make BiTEs suitable for a wide range of B-cell and autoantibody-driven neuroimmunological disorders.

## Methods

This study was conducted in accordance with all relevant ethical regulations, including compliance with the provisions of the

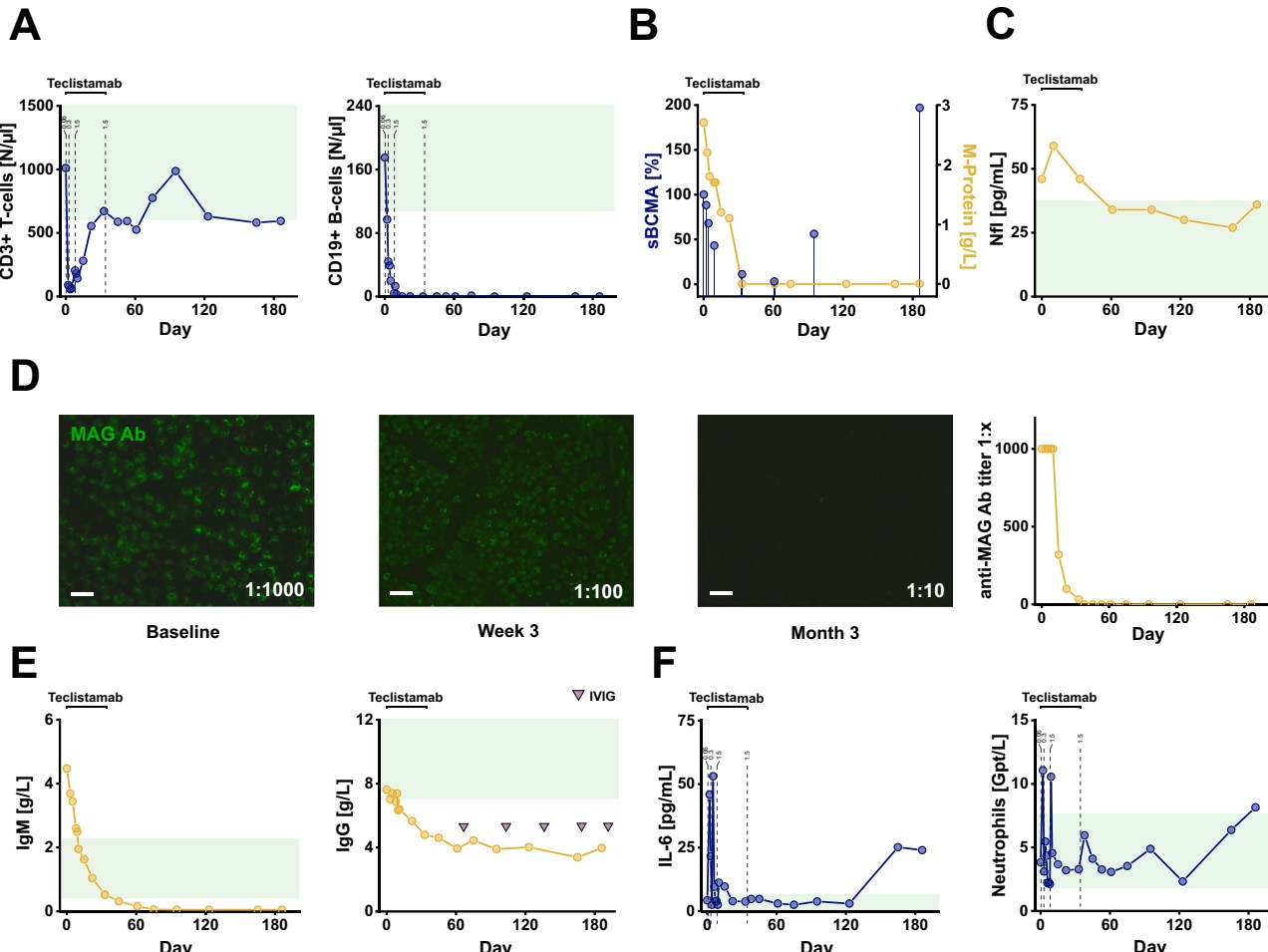

**Fig. 4 | Patient 2: Immunological changes and serologic responses in MAG neuropathy treated with teclistamab.** **A** CD3 + T-cell counts and CD19 + B-cell counts after teclistamab therapy. Green areas: Normal values. **B** Relative sBCMA levels showed a temporary decrease during the treatment phase whereas the M-protein remained undetectable until month 6. sBCMA measurements were performed three times at each time point. **C** Course of serum neurofilament (Nfl) levels showing reduction after teclistamab treatment. Green area: Normal values. **D** Left: Immunostaining of anti-MAG antibodies at baseline, 3 weeks and 3 months after first teclistamab dose showing decrease of MAG antibody titers already at week 3. MAG antibodies were negative at month 3. Right: Course of serum MAG antibody titers. Scale bars: 10 μm. **E** IgM and IgG levels after teclistamab treatment. Purple triangles indicate time points of supplementary IVIG infusion. Green area: Normal values. **F** Interleukin-6 (IL-6) levels and neutrophil levels after teclistamab. A second increase of IL-6 was observed during month 5 when the patient had a mild bronchitis. Green area: Normal values. Dashed lines in **A** and **F** indicate time points of teclistamab injections (0.06: 0.06 mg/kg body weight, 0.3: 0.3 mg/kg body weight; 1.5: 1.5 mg/kg body weight). Abbreviations: MAG-Ab: Anti-MAG antibodies, IVIG: intravenous immunoglobulin. Source data are provided as a Source Data file.

German Law. It was approved by the local ethics committee at the University Hospital Jena (Reg. No. 2024-3609-iH). 4 Patients with treatment refractory and progressive immune-mediated neuropathies were identified. Two female patients provided informed consent to a treatment with teclistamab within a compassionate use setting according to German law that permits individualized treatment approaches in cases of severe illnesses. Written informed consent for publication of their clinical details was obtained from both patients. Each patient received four doses of teclistamab (2 step up-doses of 0.06 mg/kg and 0.3 mg/kg, and 2 full treatment doses of 1.5 mg/kg) after premedication with 16 mg dexamethasone i.v., 50 mg diphenhydramin i.v. and acetaminophen 1 g p.o. and was followed up for 9 and 6 months, respectively, under initial prophylactic treatment with acyclovir (2 × 400 mg/d) and cotrimoxazole (3 × 960 mg/w). There was no commercial sponsor involved. Assessments were pre-specified and included baseline and follow-up clinical parameters, laboratory analyses, electrophysiological studies, and nerve ultrasound. For detailed methods, see Supplementary Methods.

## Flow cytometry

For flow cytometric analysis, PBMCs were thawed in a water bath at 37 °C, transferred to 50 mL Falcon tubes containing PBS + 0.5% BSA (FACS buffer) to dilute the DMSO used for freezing, and centrifuged for 5 minutes at 520 × g at 4 °C. Cells were then filtered using 70 μm cell strainers and centrifuged again for 5 min at 520 × g at 4 °C. 10 μL of Precision Count Beads (corresponding to a total of 10,300 beads) were added to the cells, which were then directly transferred to a 96-well V-bottom plate. After centrifugation for 5 minutes at 520 × g at 4 °C, the supernatant was removed and the pellets were stained in 50 μL of antibody staining mix for 30 minutes on ice. Details on the antibody staining mix can be found in Supplementary Methods. After adding 100 μL FACS buffer, cells were centrifuged for 5 min at 520 × g at 4 °C and the supernatant was removed. Cells were then resuspended in 150 μL FACS buffer and centrifuged for 5 min at 520 × g at 4 °C. After removal of the supernatant, two further washes were performed, and cells were diluted with DAPI-containing FACS buffer (1:1000), filtered through 30 μm Cell Strainer Snap Caps, and acquired using a BD FACSymphony™ S6 (laser lines: 355 nm (UV), 405 nm (violet), 488 nm

(blue), 561 nm (yellow/green), and 633 nm (red)). Data analysis was performed with FlowJo (BD Bioscience, V10). All FACS antibodies were validated by the suppliers and/or prior studies.

## Soluble B-cell maturation antigen measurements

Soluble B-cell maturation antigen (sBCMA) was measured by ELISA using a commercial kit (R&D Cat. No. DY193) alongside the corresponding reagents kits (R&D Cat. No. DY008C) according to the manufacturer's specifications. Briefly, 96-well plates were coated overnight with 100 μl capture antibody at the specified working concentration and washed three times the following day. Afterwards, plates were coated with 300 μl/well block buffer for an hour and washed again three times. Samples and standards (100 μl) were added in triplicate and washed out after 2 h. Detection antibody (100 μl) was incubated for 2 h, washed out, after which Streptavidin-HRP A (100 μl) was added to each well and incubated for 20 min. Plates were washed again with wash buffer, after which 100 μl of substrate solution was added and incubated for 20 min and the reaction was stopped with 100 μl stop solution. Optical density was measured at 450 nm with reference at 540 nm using a Spark® Multimode Microplate reader (TECAN).

## Reporting summary

Further information on research design is available in the Nature Portfolio Reporting Summary linked to this article.

## Data availability

The data generated in this study are provided in the Supplementary Information and Source Data file (main figures). All numeric data of this manuscript can also be obtained from the corresponding author upon reasonable request via email to christian.geis@med.uni-jena.de. Patient data can only be shared in pseudonymized form. Otherwise, there are no restrictions to data access. Source data are provided with this paper.

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

## Acknowledgements

We would like to thank the patients for granting their consent to publish their medical records, including their clinical history, treatment course, and related information. We thank Özge Candemir for support during the ELISA measurements. We also appreciate Heike Heske, Kathrin König, Juliane Stadtler, and Christin Reißig for their contributions to electrophysiological recordings and PBMC isolation, respectively. Additionally, we thank Katrin Hornung (Core Facility Cytometry and Institute of Immunology) for her support with the flow cytometry staining. Figures 1A and 3A were in part created in BioRender. Geis, C. (2025), https://BioRender.com/sfen2nu, https://BioRender.com/dsofstd.

## Author contributions

J.W., M.C. and C.G. designed the study. J.W., M.C., B.V., N.S., and O.Y. were involved in patient treatment. J.W. and M.C. analyzed data. D.D.

designed B-cell repertoire FACS panel, D.S. and N.R.T. established FACS panel and performed FACS analysis. J.W. and M.C. wrote the first draft of the manuscript. C.G. supervised the project. All authors discussed the results and commented on the manuscript.

## Funding

J.W., B.V., and N.S. disclose support for the research of this work from Interdisziplinäres Zentrum für Klinische Forschung Jena [ACSP13 (to J.W.), CSP 26 (to B.V.), CSP 28 (to N.S.)]. C.G. discloses support for the research of this work from German Research Foundation – Deutsche Forschungsgemeinschaft [GE2519/8-2 and GE2519/9-2], the German Federal Ministry of Research, Technology and Space [01GM1908E] and Schilling foundation. D.D. discloses support for the research of this work from German Research Foundation – Deutsche Forschungsgemeinschaft [DU548/9-1 (515982377)]. D.D. is principle investigator and N.R.T. is a postdoctoral fellow funded by the DFG under Germany's Excellence Strategy – EXC 2051 – Project-ID 390713860. M.C., D.S. and O.Y. declare no relevant funding. Open Access funding enabled and organized by Projekt DEAL.

## Competing interests

The authors declare the following competing interests: J.W. reports honoraria from Argenx, Alexion, travel support from Neuraxpharm, and research support from Ionis Pharmaceuticals. M.C. reports honoraria from Argenx. C.G. received speaker fees and compensation for advisory boards from Alexion, Roche, Sobi, Argenx, Kyowa, Astra Zeneca and travel support from Alexion. D.D. received compensation for advisory board from Johnson & Johnson and received funding from the company Affimed in an independent project. B.V., N.S., D.S., N.R.T., R.T., and O.Y. declare no competing interests.
