## [Transparent Peer Review file · Nature Communications]

Therapeutic effect of T-cell engager in two patients with autoimmune neuropathy

Corresponding Author: Professor Christian Geis

Version 1:

Reviewer comments:

Reviewer #3

(Remarks to the Author)

Reviewer #4

(Remarks to the Author)

The manuscript by Wickel and colleagues is the first to describe the effects of the T cell engager (TCE) teclistmab (a CD3 – BCMA bispecific antibody leading to B cell and plasma cell depletion) on two patients with autoimmune, paraproteinemic neuropathy. The manuscript has been transferred from Nat Medicine to Nat Commun and was initially reviewed by two other reviewers. The authors now provide a revised manuscript and a comprehensive reply to the reviewers' suggestions. This is an innovative study providing a thorough assessment of teclistmab effects on the patients' clinical presentation, electrophysiological and sonography findings, and extended laboratory analysis. In the revised version, the authors present additional data that further substantiate the rationale and help elucidate the mechanisms underlying the rapid and profound therapeutic effect.

The previous comments of the reviewers comprised (1) to provide more information on preceding therapies, (2) short duration of follow-up, (3) lack of mechanistic data to substantiate a disease reset, (4) missing references.

(1) The authors now include a detailed overview on previous disease dynamics and (unsuccessful) therapies in the cases with IgM-kappa paraprotein and MAG antibody associated neuropathy. In a new supplementary table they provide information that standard therapies did not lead to significant and sustained improvement despite application of methylprednisolone, plasma exchange, IvIg, and rituximab.

(2) Information on follow-up duration is now shown until 9 and 6 months for patient one and two, respectively. The author claim that these observation periods are in line with the follow-up periods in most phase 3 studies and current reports on TCE and CAR T cell therapies in autoimmune disorders – which seems reasonable. It is worth mentioning that during this extended observation period, the clinical presentation even improved further and there is no evidence for relapse or secondary deterioration.

(3) The authors acknowledge that reporting mechanistic data in individual case reports is somewhat limited but they now include detailed flow cytometry analysis of one patient comparing the repopulated B cell fraction with baseline values showing a more naïve phenotype of the newly emerged peripheral B cells (in patient two B cells were still depleted). Interestingly, although no bone marrow biopsies were performed, authors were able to show a temporary decline of soluble BCMA as a proxy for plasma cell expansion in both patients. Corroborating the conclusion of immune reconstitution by naïve B cells, the pathogenic antibodies and M gradient were absent until the end of the observation period in both patients. They furthermore provide information on reduced neuronal damage by measuring NFL, which fits in line of marked improvements in neurophysiology.

(4) Relevant references have been added, including citations on concurrent reports on use of TCE and CAR T cells in immune-mediated disorders.

Therefore, in my view, the previous reviewers' comments are indeed fully addressed and authors also acknowledge limitations which are unavoidable in a compassionate use setting.

In summary, this is a timely and innovative contribution to the new and emerging therapies in neuroimmunology. The data is

comprehensive and convincing and the rationale of treating therapy-resistant paraproteinemic neuropathies with plasma-cell directed therapy is plausible and obviously highly effective. There is no doubt that TCE will become a highly effective treatment option in immune-mediated disorders in rheumatology and neurology. Thus, this report represents a meaningful advance, merits publication, and is likely to attract significant interest in the field.

We thank the reviewers for their positive feedback on the manuscript. There were no additional points that needed to be addressed.

REVIEWERS' COMMENTS

Reviewer #4 (Remarks to the Author):

The manuscript by Wickel and colleagues is the first to describe the effects of the T cell engager (TCE) teclistmab (a CD3 – BCMA bispecific antibody leading to B cell and plasma cell depletion) on two patients with autoimmune, paraproteinemic neuropathy. The manuscript has been transferred from Nat Medicine to Nat Commun and was initially reviewed by two other reviewers. The authors now provide a revised manuscript and a comprehensive reply to the reviewers' suggestions.

This is an innovative study providing a thorough assessment of teclistamab effects on the patients' clinical presentation, electrophysiological and sonography findings, and extended laboratory analysis. In the revised version, the authors present additional data that further substantiate the rationale and help elucidate the mechanisms underlying the rapid and profound therapeutic effect.

The previous comments of the reviewers comprised (1) to provide more information on preceding therapies, (2) short duration of follow-up, (3) lack of mechanistic data to substantiate a disease reset, (4) missing references.

(1) The authors now include a detailed overview on previous disease dynamics and (unsuccessful) therapies in the cases with IgM-kappa paraprotein and MAG antibody associated neuropathy. In a new supplementary table they provide information that standard therapies did not lead to significant and sustained improvement despite application of methylprednisolone, plasma exchange, IVIg, and rituximab.

(2) Information on follow-up duration is now shown until 9 and 6 months for patient one and two, respectively. The author claim that these observation periods are in line with the follow-up periods in most phase 3 studies and current reports on TCE and CAR T cell therapies in autoimmune disorders – which seems reasonable. It is worth mentioning that during this extended observation period, the clinical presentation even improved further and there is no evidence for relapse or secondary deterioration.

(3) The authors acknowledge that reporting mechanistic data in individual case reports is somewhat limited but they now include detailed flow cytometry analysis of one patient comparing the repopulated B cell fraction with baseline values showing a more naïve phenotype of the newly emerged peripheral B cells (in patient two B cells were still depleted). Interestingly, although no bone marrow biopsies were performed, authors were able to show a temporary decline of soluble BCMA as a proxy for plasma cell expansion in both patients. Corroborating the conclusion of immune reconstitution by naïve B cells, the pathogenic antibodies and M gradient were absent until the end of the observation period in both patients. They furthermore provide information on reduced neuronal damage by measuring NFL, which fits in line of marked improvements in neurophysiology.

(4) Relevant references have been added, including citations on concurrent reports on use of TCE and CAR T cells in immune-mediated disorders.

Therefore, in my view, the previous reviewers' comments are indeed fully addressed and authors also acknowledge limitations which are unavoidable a compassionate use setting.

In summary, this is a timely and innovative contribution to the new and emerging therapies in neuroimmunology. The data is comprehensive and convincing and the rationale of treating therapy-

resistant paraproteinemic neuropathies with plasma-cell directed therapy is plausible and obviously highly effective. There is no doubt that TCE will become a highly effective treatment option in immune-mediated disorders in rheumatology and neurology. Thus, this report represents a meaningful advance, merits publication, and is likely to attract significant interest in the field.